# Effects of Task Constraints on Kinematic Characteristics of Body Segments in Children with Cerebral Palsy and Children with Typical Development during Hand-Reaching

**Yun-Huei Ju [1,2,\*] and Rong-Ju Cherng [3,4]**

1   Department of Physical Therapy, College of Health Science, Kaohsiung Medical University,
    Kaohsiung 807378, Taiwan
2   Department of Rehabilitation Medicine, Kaohsiung Medical University Hospital, Kaohsiung 807378, Taiwan
3   Institute of Allied Health Sciences, National Cheng Kung University, Tainan 701401, Taiwan;
    rjc47@mail.ncku.edu.tw
4   Department of Physical Therapy, National Cheng Kung University, Tainan 701401, Taiwan
\*   Correspondence: yuhujw@kmu.edu.tw

**Abstract:** Background: Children with cerebral palsy (CP) have difficulty in managing postural control during functional reaching tasks, although children with different postural control ability are able to come up with different motor solutions to cope with different task demands. This study examined the effect of task constraint on postural control performance in children with cerebral palsy and typical development (TD) in terms of different postural control abilities. Methods: A cross-sectional research design was used. Twelve children with spastic diplegic cerebral palsy (mean age: 107.8 months) and 16 typically developing children (mean age: 110.9 months) participated in this study. Individually, all subjects were seated in a height-adjusted chair and were requested to reach for target(s) located at three different directions (medial, anterior, and lateral). A six-camera Qualisys Motion Capture System was used to capture motion data. Kinematic data in terms of body alignment and angular changes were analyzed. Results: Children with cerebral palsy demonstrated different postural control strategies to complete different reaching tasks compared to typically developing children by preparing postural alignment in advance, coordinating different body orientation movements during reaching after showing difficulty in managing reach medially. Conclusions: Children with cerebral palsy perceive their insufficient ability and prepare their alignment in advance to adapt to the task demanded and decrease the postural challenges of the task. Even though children with cerebral palsy self-generate different motor solutions to reach without falling, these alternative strategies might not be the most efficient adaptation.

**Keywords:** postural control; cerebral palsy; reach; task constraint

## 1. Introduction

Task context could constrain the output of motor behavior in terms of movement patterns [1–3]. Motor behavior is developed from the interaction amongst task, individual, and environment constraints [4]. Individuals would actively perceive these features and choose preferred motor patterns (motor solutions) to achieve task goals. These preferred motor patterns should fit with the task demands [4]; however, these preferred motor patterns are not necessarily consistent across groups characterized with different features such as children with cerebral palsy (CP) vs. typical development (TD) [5].

Children with CP are characterized with motor difficulty and insufficient postural control, which limit children's normal developmental functional activity [6,7]. The literature presents children with CP as demonstrating stereotypical motor control, inadequate coordination, motor inefficiency, slow speed of movement, use of immature motor synergy, use of variable recruitment patterns, delayed response with external disturbance, insufficiency of subtle motor adjustment/regulation, and insufficient anticipated postural control,

etc., as they perform functional tasks or deal with balance threats [8–14]. These motor problems could be affected by these multiple factors, and as children with CP have deficit in sensorimotor integration, anticipating control of force regulation could be impaired [15]. Rose and McGill found that children with CP encountered difficulty in recruiting motor units during force regulation [16]. Consequently, children with CP might use alternative motor strategies to complete functional tasks.

Several studies have found that severity of involvement and biomechanical constraints play important roles in the motor control of children with CP [14,17,18]. Due to the specific characteristics of this group, it appears essential for PT clinicians to recognize how children with CP react to tasks considering their specific postural control ability. Such information would be very valuable in clinical settings when attempts are made to mediate their problems with performing tasks.

The seated reaching task is commonly seen in daily activities, such as reaching for a book sitting in front of a table or passing over a report to the next person (rotating the trunk and reorienting the body in 3D space in order to let the upper arm reach for a specific location). Since children with CP may have more problems with standing; it could be easier for them to sit and manage tasks, as pointed out by Westcott and Burtner (2004) [13], although seated reaching tasks have not yet been thoroughly studied.

Several studies regarding reaching tasks have been reported [11,12,19]. Van der Heide et al. (2004) found that most children (CP and TD groups, aged 2–11 years) demonstrated direction-specific postural control in reaching; however, they still observed that some children (aged 8–11) with CP demonstrated variable motor performance, and anticipatory postural control was seldom observed among them [20]. They also found that children with CP seemed to develop at least three control patterns of postural adjustment: (1) variable patterns of postural control, (2) stereotyped patterns of weakly modulating postural control, and (3) stereotyped patterns with stiff postural control, implying that these children might balance their body according to previous experience for reaching tasks. Additionally, they discovered that selective postural control, in accordance with the balancing situation, could be the key component for children with CP functioning well in daily life. Other researchers have found children with CP could balance themselves with specific muscle recruitment at different balancing conditions, demonstrating higher levels of self-care performance than children who could not react selectively in such situations. This evidence supports a relationship between postural control ability and upper extremity functionality.

In another study, researchers examined the reach performance (kinematic data) of people with hemiplegic CP (age: 11–27 years old) compared to a control group with typical development (age-matched) [21]. Impaired forearm supination was characterized in the CP group. The results showed that people with CP recruited extra-trunk movement to compensate for their deficit of upper extremity movement; additionally, yet another research study showed similar findings [22]. Children with CP and TD (aged 5–14 years) performed a reach-and-drink task with supported seated posture, and children with CP increased forward trunk motion compared to children with TD while conducting the task [22]; however, this extra trunk displacement of children with CP occurred in a supported sitting position. In Robert and Levin's study (2018), the researchers also found a similar result, as they requested children with hemiplegic CP to reach targets on the frontal plane and found that this specific group demonstrated less trunk rotation than TD children during reach [23]. It seems that people with CP perceive their characteristics and develop a motor solution (alternative movement pattern) to complete functional tasks. Kreulen et al. derived a detailed kinematic analysis as indispensable in planning therapy for people with motor difficulty [21].

Although these studies have revealed that children with CP demonstrated several motor control problems during reaching, the relationship between postural control ability and hand-reach performance was not consistently tested. The children were not seated freely (certain postural constraints were used in these studies) while reaching for a target; additionally, previous researchers might have recruited children with hemiplegic CP who

had less postural control disadvantage compared to children with diplegic CP [23]. Accordingly, limited kinematic information regarding reaching performance in children with diplegic CP has been reported, such as head, trunk, and pelvis alignment and movement since the subjects have difficulty managing to balance themselves during functional tasks.

Formerly, our research revealed that the outcome of hand performance (movement units and straightness ratio) in terms of coordination and smoothness of movement is significantly correlated with postural control ability by children with CP [24]. The children with insufficient postural control ability produced inadequate coordinated and inefficient hand reach. In addition, children with CP demonstrated different patterns of postural adjustment in terms of center of pressure (COP) performance compared to TD children when reaching in different directions [25]. The findings of this study revealed that COP performance (e.g., sway ratio: SR) of children was negatively correlated with hand movement units (MUs). Children with insufficient postural control ability (low PRT score) also showed wide COP excursion (small SR value) with the evidence of increasing absolute peak COP velocity in the medial–lateral direction. This means children with CP, demonstrating poor postural control ability (low PRT score), showed big medial–lateral sway (small SR value) and also demonstrated poor coordinated hand reach motion (increased MUs) during reach out.

Our previous research reported that children with wide COP excursion in reaching tasks seemed to produce great hand-movement units (MUs) [25]. In other words, children with less postural control ability demonstrated less coordinated hand-reach performance. Furthermore, children with CP reacted differently than TD children when reaching medially (deviated 40 degrees medially from anterior reach) and laterally (deviated 40 degrees laterally from anterior reach). Increased SR in both lateral and medial reach was observed in children with CP. A rather stable SR value (about 1) was shown in children with TD. Children with CP become unstable during reaching laterally and medially due to increased balancing threats.

Two of our research results have been reported in terms of hand-reach performance and effect of postural control ability on reach performance [24,25]. For this research report, in order to examine the coordination between hand reach and postural control in terms of kinematic performance, we analyzed kinematic data as children participants performed a seated reach task without any external postural support. The purpose of this research report was specifically to examine the effect of the task constraint on postural control performance in children with CP and TD (in terms of different postural control abilities). We hypothesized that children with CP should perform different forms of angular patterns of head, trunk, and pelvis segments to manage different tasks than TD children because of the different postural control ability between them.

## 2. Materials and Methods

### 2.1. Participants

Twelve children with spastic diplegic CP and 16 TD children participated in this study. A convenience sample was used. No significant differences were found in these two groups regarding age (CP: 107.8 months vs. TD: 110.9 months; $p = 0.723$) body weight (CP: 26.7 kg vs. TD: 30.3 kg; $p = 0.170$), or gender distribution (CP: 5/7 (F/M) vs. TD: 10/6, $p = 0.274$). The average seated height of children with CP (92.7 cm) was significantly shorter than that (102 cm) of TD children ($p = 0.005$). Children with CP demonstrated a gross motor function level ranging from level II to IV based on the Gross Function Measure Classification System (GFMCS) [26]. For details of characteristics of children with CP please refer to a previous study [25]. All the parents/guardians of the participants signed informed consents, and the participants gave their oral assent. The protocol of the study was approved by the Institutional Review Board of Kaohsiung Medical University, Chung-Ho Memorial Hospital, and an ethical review was carried out (KMUH-IRB-930057).

### 2.2. Experimental Task

All children were asked to sit on a stool with an adjustment for his/her seated height. Each child reached for a puppy target with a distance of 120% of his/her arm length. Three reach directions were set (ant task, lat40 task, and med40 task): (1) anterior to the dominant (or preferred) shoulder (ant task), (2) deviated 40 degrees laterally from the dominant (or preferred) shoulder (lat40 task), and (3) deviated 40 degrees medially from the dominant (or preferred) shoulder (med40 task). All children reached for the target and returned to the seated position with his/her preferred hand with certain reach speed regulated by a metronome. Detailed information was reported in our previous study [24,25].

### 2.3. Clinical Measure

All participants received anthropometric measurements including arm length, knee and ankle width, body weight, and seated height. They were also asked to throw a ball three times to confirm their dominant (preferred) hand. The ability of postural control was tested with the Pediatric Reach Test (PRT) proposed by Bartlett and Birmingham in 2003 [27]. For the PRT, each child was requested to reach out as far as he/she could, then the difference of the distance (cm) between the start and the farthest position of the hand was documented, with three reach directions (anterior, bilateral) in two postures (sitting and standing) conducted. Three scores—a sitting score (distance: cm), a standing score, and a total score (sum of sitting and standing score)—were then derived.

In addition, the Gross Motor Functional Classification System (GMFCS) was used to classify the extent of gross motor function level for children with CP [26]. The sit and stand dimensions of the Gross Motor Function Measure (GMFM) were conducted to measure the functional ability of children with CP [28] in order to represent the certain degree of a child's postural control ability from functional perspectives. The interrater reliability of PRT and GMFM was determined and reached above 0.90 in this study.

### 2.4. Equipment and Setup

A six-camera Oqus 100 series Motion Capture System (Qualisys Motion Capture Systems, Qualisys AB, Sweden) with a 150 Hz sampling rate recorded the positions of reflective markers. Reflective markers were attached to the following body landmarks: the point between the eyebrows, the bilateral mastoid process of the occipital region T2, the bilateral acromion process, the bilateral medial/lateral epicondyles, the bilateral medial and lateral styloids of the forearm, the bilateral 3rd metacarpal head of hands, the bilateral anterior superior iliac spine (ASIS), the midpoint between the bilateral posterior superior iliac spine (PSIS), the midpoints of the bilateral thigh, the bilateral lateral epicondyle of the femur, the midpoints of the lower legs, the bilateral lateral malleolus, and triads (three reflective markers) on the upper arms and forearms.

The puppy target was fixed in a container hung from a ceiling. The height and distance of the target were adjusted to each child's seat height and arm length. Two Kistler force platforms with 150 Hz measured the ground reaction force (GRF) and center of pressure. Three digital videos (DV) were used to record the children's reach performance. One force plate was placed underneath the stool and the other was placed underneath the feet.

### 2.5. Procedure

Before the lab testing, all children were first asked to throw a ball three times to confirm the side of the dominant hand (which also indicated the lateral hand), then clinical measures were applied.

All the children were seated on a stool with a posture of hips and knees in 90–90 degrees and feet flat on a force plate without back or arm support. The starting position of the dominant (or preferred) hand was put on the children's lap, marked with a sticker to indicate where to return to. The non-dominant hand (medial hand) was positioned along the trunk and relaxed. A verbal signal was given to indicate initiation of reach and return.

Before the record of reaching tasks, neutral positions of each body segment were measured in order to define the 0 degree of each angle. Children were also asked to sit in the starting position.

After all the preparation, the children were asked to perform the reach–return task. All equipment was synchronized and recorded the reach–return performances, with the pace of the reach-and-return task modulated with a metronome (46 beats/min). Children were allowed to practice several times to get used to the pace, then each child performed six trials for each condition. In total, 18 trials were conducted. The order for the task was randomly assigned.

*2.6. Data Reduction*

Qualisys Tracker Manager (QTM) software was used to convert the positioned raw data from the Oqus cameras into three-dimensional coordinates and to calculate force and center of pressure (COP) data from two force platforms. The videos were used to check reaching–returning movements with the QTM program while defining events of reach begin, target touch, target leave, and return end. The position raw data were filtered with a filter with cutoff frequency of 5 Hz. Hand reach begin was defined as a continuum of changes of hand velocity above 5% of peak velocity. A return end was defined as a continuum of changes of hand velocity below 5% of peak velocity. Touch target and leave target events were defined by 5-volt signals (signal on/off) as hand touch and target leave.

Angular displacement was calculated. For angular motion, a local system for each body segment was established with attached reflective markers. Euler's method, with an y–x–z rotational sequence, was used to calculate the angular motion for each joint (head, shoulder, elbow, wrist, upper trunk, pelvis, hip, knee, and ankle). For angular data set averaging, they were standardized with 100 points of the data set and aligned with the reach-begin event.

Postural alignment 300 ms before the preferred hand started to reach was calculated and averaged with 300 ms of the period of time before hand movement. This was used to indicate postural preparation before task performance. Angular changes in the head, upper trunk, pelvis, hips, knees, and ankles were calculated at the hand acceleration (the time period between hand begin and hand peak velocity) and deceleration phases (the time period of hand peak velocity and target touch).

*2.7. Statistical Analysis*

For each of the above-mentioned variables, the average of three trials for each participant and each direction was used in the statistical analyses. The primary analyses were MANOVA with the group as the between-subject factor and reaching directions as the within-subject factors. Subsequent post hoc multiple comparisons were used with the LSD procedure. The significance level was set at 0.05.

## 3. Results

*3.1. Mean Alignment 300 ms before Preferred Hand Move during Reach Out*

Group effects existed in the transverse plane (rotation) of head and trunk movement (head: F = 4.816, df = 1, $p$ = 0.009; trunk: F = 3.080, df = 1, $p$ = 0.046) and in the frontal plane of pelvis movement (F = 5.995, df = 1, $p$ = 0.003). Children with CP apparently aligned their head with greater degrees of rotation in advance than did TD children (Table 1), indicating that children with CP prepared their body orientation with consideration of the reaching direction. For the task effect, it was found that children side-bent their head laterally before anterior and lateral reaches, and changed head position back toward neutral before initiating medial reach (F = 9.555, df = 2, $p$ = 0.001).

Children with CP reacted differently than TD children in different task directions in terms of the transverse plane of head and trunk alignments by apparently adjusting their head rotation alignment more laterally or medially before reaching in accordance with the task directions compared with TD children, especially when reaching medially

(post hoc multiple comparisons with LSD procedure: CP mean: $-7.21°$ vs. TD mean: $-0.07°$, $p = 0.027$). For the interaction effect of trunk rotation (F = 4.092, df = 2, $p = 0.029$), children with CP positioned their trunk with greater degrees of rotation to the lateral side than TD children before anterior and lateral reaches. Children with CP aligned the trunk with certain degrees of rotation toward the medial side before medial reach, whereas TD children more likely kept the trunk in a neutral position before reaching.

**Table 1.** Mean and standard deviation of dependent variables of postural alignment 300 ms before the preferred hand started to reach.

| Task | CP | | | TD | | |
|---|---|---|---|---|---|---|
| Variable (°) | ant | lat40 | med40 | ant | lat40 | med40 |
| Head extension/flexion | 4.83 (10.52) | 4.55 (7.78) | 5.30 (9.17) | 0.56 (8.55) | 2.45 (9.72) | 0.85 (9.72) |
| Head side bend to lat/med | 5.16 (10.20) | 6.47 (9.61) | −0.74 (8.03) | 2.58 (3.13) | 2.95 (5.54) | 0.05 (2.95) |
| Head rotation to lat/med | 6.66 (6.13) | 10.17 (10.88) | −7.21 (10.36) | 4.21 (4.49) | 6.40 (7.93) | −0.07 (6.03) |
| Trunk extension/flexion | −2.95 (14.44) | −2.73 (15.01) | −3.58 (15.24) | −0.55 (9.31) | 0.06 (8.18) | 1.38 (9.95) |
| Trunk side bend to lat/med | −1.70 (2.89) | −1.56 (3.70) | 0.28 (4.72) | −2.90 (2.58) | −3.15 (2.67) | −1.94 (2.63) |
| Trunk rotation to lat/med | 3.26 (5.31) | 4.23 (5.09) | −1.12 (3.78) | −0.29 (2.42) | 1.26 (1.76) | −0.71 (3.35) |
| Pelvis post/ant tilt | −1.05 (3.46) | 0.48 (2.29) | 0.41 (2.93) | 0.42 (1.21) | 0.34 (0.78) | 0.81 (0.65) |
| Pelvis tilt down to lat/med | −1.85 (5.38) | −4.01 (10.54) | −2.53 (2.51) | −0.06 (0.52) | −0.22 (0.65) | −0.14 (0.69) |
| Pelvis rotation to lat/med | −0.67 (1.81) | −0.37 (2.30) | −0.60 (1.22) | 0.05 (0.90) | 0.25 (0.98) | −0.29 (1.07) |
| Lateral hip flexion/extension | 89.61 (14.32) | 90.05 (15.32) | 89.03 (13.82) | 63.06 (7.57) | 63.70 (6.96) | 63.54 (8.86) |
| Lateral hip adduction/abduction | 2.30 (8.22) | 3.65 (10.49) | 2.52 (6.46) | −10.14 (6.75) | −10.80 (7.87) | −11.36 (7.62) |
| Medial hip flexion/extension | 89.27 (15.81) | 89.88 (16.66) | 90.06 (14.55) | 66.32 (5.50) | 66.31 (5.92) | 66.48 (6.37) |
| Medial hip adduction/abduction | 2.78 (12.53) | 2.02 (12.97) | −0.11 (11.75) | −4.87 (6.97) | −4.68 (5.86) | −3.76 (5.86) |
| Lateral ankle dorsi/plantarflexion | 0.37 (6.15) | 1.17 (9.23) | 1.53 (8.08) | 5.68 (13.00) | 4.32 (13.65) | 3.85 (12.62) |
| Lateral ankle supination/pronation | −5.13 (10.49) | −5.61 (10.76) | −6.13 (8.57) | 10.42 (13.49) | 9.49 (12.83) | 9.80 (12.67) |
| Lateral ankle abduction/adduction | −4.62 (6.98) | −3.71 (9.02) | −3.07 (8.20) | 30.15 (10.95) | 30.17 (11.41) | 29.15 (11.23) |
| Medial ankle dorsi/plantarflexion | −5.48 (7.72) | −4.35 (5.88) | −4.38 (6.60) | −2.96 (9.75) | −2.78 (9.43) | −2.06 (9.12) |
| Medial ankle supination/pronation | −2.14 (10.09) | 0.71 (9.00) | 0.32 (9.16) | −8.85 (8.84) | −10.42 (9.33) | −10.85 (9.64) |
| Medial ankle abduction/adduction | 0.51 (5.86) | −1.40 (7.35) | −0.53 (7.10) | 19.64 (10.65) | 20.39 (9.61) | 18.73 (10.54) |

Note: lat: lateral side (the same side as lateral reach; the same side as the preferred hand); med: medial side (the same side as medial reach; the same side as non-preferred hand.). The movement direction first mentioned in the variable is valued as being positive; for example, head extension is valued as positive.

For pelvis movement, children with CP tended to position their pelvis tilt more to the medial side before reaching than TD children (significant group effect). No task or interaction effect of the pelvis was found.

A group effect of the hips was found in hip alignment 300 ms before reaching. Children with CP aligned their hip in a more flexed and adducted position than did TD children (flexion: F = 13.732 and 11.752, respectively, df = 1, $p < 0.001$; adduction/abduction: F = 8.416 and 3.045, respectively, df = 1, $p < 0.001$ and $p = 0.047$, respectively). An interaction effect of the medial hip was found in adduction/abduction movement (F = 4.267, df = 2, $p = 0.025$). Before anterior and lateral reaches, children with CP aligned the medial hip in an adduction position (pointing toward the reach direction), but TD children did this in abduction position (away from the reach direction). At medial reach, children with CP adjusted the hip in a rather neutral position, but TD children used the hip abduction position.

The group effect of ankle alignment 300 ms before the reach was significant, except for the alignment of ankle dorsiflexion and plantarflexion (F = 0.985 and F = 0.220, respectively,

df = 1, *p* = 0.416 and *p* = 0.882, respectively). Children with CP aligned the lateral ankle in a pronation and adduction (positioned foot inward) position, whereas TD children aligned the lateral ankle in a supination and abduction (positioned foot outward) positions (Table 1). Children with CP aligned the medial ankle in a rather neutral position, whereas TD children aligned the medial ankle in a pronation and abduction position. Neither task nor interaction effect was seen.

In summary, children with CP were aware of the different reach directions and decided to align their head and trunk with certain degrees of rotation in accordance with the reach directions in advance, thereby orienting their body in advance before reaching. For the lower extremities, children with CP positioned their legs with the knees closed, but TD children kept the knees apart. It was also found that children with CP likely adjusted the medial hip position before reaching.

### 3.2. Angular Changes in Axial and Lower Extremity Components during the Hand Acceleration Phase

For head movement, a group effect existed in the head rotation direction (F = 4.709, df = 1, *p* = 0.010). Children with CP rotated their head to the medial side at anterior reach, whereas TD children rotated it to the lateral side (Table 2); noticeably, children with CP rotated their head to a greater degree than did the TD children. Task effects were found in head extension/flexion (F = 13.171, df = 2, *p* < 0.001) and rotation movements (F = 23.636, df = 2, *p* < 0.001). In general, all children rotated their head with the greatest degree to the lateral side at lateral reach. No interaction effect of head movement between groups or tasks was found during the hand acceleration phase.

**Table 2.** Mean and standard deviation of dependent variables of angular changes in axial and lower extremity components during the hand acceleration phase.

| Task / Variable (°) | CP | | | TD | | |
|---|---|---|---|---|---|---|
| | ant | lat40 | med40 | ant | lat40 | med40 |
| Head extension/flexion | −0.73 (2.23) | 1.57 (4.33) | −5.37 (4.89) | −0.99 (1.31) | −0.48 (2.85) | −2.99 (2.73) |
| Head side bend to lat/med | −0.10 (2.52) | 1.23 (4.08) | 0.66 (2.32) | 1.54 (1.41) | 1.44 (2.42) | 1.75 (2.13) |
| Head rotation to lat/med | −0.01 (2.72) | 7.46 (4.00) | −3.74 (4.82) | 3.99 (2.97) | 9.96 (7.24) | −0.021 (4.42) |
| Trunk extension/flexion | 2.27 (2.31) | 1.04 (2.43) | 7.15 (4.19) | 4.00 (2.50) | 2.64 (1.92) | 4.94 (3.35) |
| Trunk side bend to lat/med | −0.02 (1.43) | −0.20 (2.66) | −1.72 (2.43) | −0.91 (1.51) | −0.94 (1.20) | −0.15 (3.31) |
| Trunk rotation to lat/med | −3.48 (1.98) | −1.29 (1.76) | −7.63 (4.83) | −5.34 (2.62) | −0.04 (2.58) | −10.04 (3.94) |
| Pelvis post/ant tilt | −0.59 (1.62) | −1.27 (1.85) | −1.34 (2.29) | −1.00 (1.10) | −0.72 (0.97) | −0.58 (1.22) |
| Pelvis tilt down to lat/med | −0.17 (1.09) | −0.31 (1.01) | −0.94 *1.99) | −0.11 (0.81) | −0.00 (0.64) | −0.09 (1.00) |
| Pelvis rotation to lat/med | 0.88 (0.68) | 0.67 (1.59) | −0.47 (2.15) | −0.10 (0.84) | −0.04 (0.86) | −2.46 (3.14) |
| Lateral hip flexion/extension | 1.43 (2.06) | 1.83 (2.49) | 2.27 (2.86) | 1.65 (1.41) | 1.26 (1.16) | 0.61 (1.92) |
| Lateral hip adduction/abduction | 0.01 (2.71) | −0.79 (3.03) | 1.44 (1.59) | −0.45 (0.69) | −0.22 (1.11) | −2.31 (4.01) |
| Medial hip flexion/extension | 1.09 (2.04) | 1.27 (2.78) | 1.09 (3.50) | 1.18 (1.11) | 0.93 (0.72) | 0.74 (1.31) |
| Medial hip adduction/abduction | −0.02 (1.91) | −0.25 (1.91) | −0.10 (1.67) | 0.66 (0.65) | 0.26 (0.83) | 1.58 (1.09) |
| Lateral ankle dorsi/plantarflexion | 0.14 (1.48) | 0.48 (1.34) | −0.38 (3.47) | −0.37 (0.86) | 0.08 (0.36) | −0.01 (1.53) |
| Lateral ankle supination/pronation | 0.39 (1.36) | 1.25 (1.99) | −0.99 (3.48) | −0.15 (0.83) | 0.16 (0.59) | −0.26 (1.36) |
| Lateral ankle abduction/adduction | −0.33 (2.07) | −0.33 (2.75) | 0.36 (3.07) | −0.72 (1.37) | −0.07 (0.62) | −0.55 (1.50) |
| Medial ankle dorsi/plantarflexion | 0.17 (1.16) | 0.34 (0.84) | −0.88 (1.39) | −0.20 (0.27) | −0.02 (0.32) | 0.42 (1.25) |
| Medial ankle supination/pronation | 0.93 (3.34) | 0.10 (1.14) | −0.36 (2.74) | −0.27 (0.23) | −0.06 (0.32) | −1.05 (1.91) |
| Medial ankle abduction/adduction | −0.01 (1.27) | 0.41 (1.39) | 1.14 (2.09) | 0.17 (0.32) | −0.19 (0.38) | −0.94 (2.63) |

Note: lat: lateral side (the same side as lateral reach; the same side as the preferred hand); med: medial side (the same side as medial reach; the same side as the non-preferred hand).

For trunk movement, group, task, and interaction effects were found, except in certain directions of trunk movement. Children with CP extended the trunk with fewer degrees relative to the pelvis than did TD children at anterior and lateral reaching; however, children with CP increased the degree of trunk extension at medial reach. For the trunk side-bending movement, the two groups of children showed different movement adjustments at three reach directions (interaction effect: $F = 3.324$, df = 2, $p = 0.052$). Children with CP showed noticeable trunk side-bending to the medial side at medial reach, but showed no significant movement at anterior or lateral reaches. TD children showed subtle trunk side-bending to the medial side at anterior and lateral reaches, but showed no trunk side-bending movement at medial reaches.

For trunk rotation, children demonstrated a specific direction of trunk rotation for reach in different directions. Basically, they could increase the range of trunk rotation to the medial side at medial reach and decrease the range of trunk rotation to the medial side at lateral reach (task effect: $F = 46.774$, df = 2, $p < 0.001$). Furthermore, children with CP and TD both demonstrated different angular adjustments of trunk rotation (interaction effect: $F = 4.240$, df = 2, $p = 0.025$). TD children showed several degrees of trunk rotation to the medial side for anterior reach, and then performed no trunk rotation for lateral reach. They showed the greatest degree of trunk rotation for medial reach (mean of anterior: $-5.34°$, mean of lateral: $-0.04°$, and mean of medial: $-10.04°$). Children with CP did not show this apparent adjustment of trunk rotation (mean of anterior: $-3.48°$, mean of lateral: $-1.29°$, and mean of medial: $-7.63°$).

For pelvis movement, group ($F = 7.187$, df = 1, $p = 0.001$) and task effects ($F = 4.925$, df = 2, $p = 0.015$) existed in pelvis rotation. Children with CP rotated the pelvis toward the lateral side for anterior and lateral reaches, whereas TD children did not. At medial reach, children with CP did not demonstrate significant pelvis rotation to the medial side, as did TD children. A significant interactive effect was found in the pelvis tilting down to the lateral/medial sides ($F = 3.680$, df = 2, $p = 0.039$). Children with CP showed different directions of pelvis tilting (down to lateral or medial side) when reaching in different directions, especially when reaching medially. TD children maintained stable pelvis movement while reaching (Table 2).

For hip movement, a group effect existed in hip adduction/abduction ($F = 4.028$ and $F = 4.512$, respectively, df = 1, $p = 0.018$ and $0.012$, respectively). Children with CP showed certain degrees of lateral hip adduction and no movement of the medial hip (swaying the lower extremity toward the medial side of the trunk), whereas TD children showed certain degrees of lateral hip abduction and certain degrees of medial hip adduction (swaying the lower extremity away from the medial side of the trunk) during medial reach. An interaction effect between group and task did exist in the lateral hip adduction/abduction ($F = 4.996$, df = 2, $p = 0.015$), where children with CP adducted the lateral hip at medial reach, but TD children tended to abduct the lateral hip at medial reach.

For ankle movement, a group effect was revealed in lateral ankle supination/pronation ($F = 2.928$, df = 1, $p = 0.053$). It seemed that children with CP demonstrated less stability of ankle motion (apparent ankle motion) than TD children (mild ankle motion) during the hand acceleration phase (Table 2). A task effect existed in medial ankle supination/pronation ($F = 4.312$, df = 2, $p = 0.024$), whereas an interaction effect was seen in medial ankle dorsiflexion/plantarflexion ($F = 5.609$, df = 2, $p = 0.009$). During lateral reach, children with CP supinated the lateral ankle, but TD children did not. For the task effect, children likely tried to pronate the medial ankle more during medial reach than during anterior reach. Interestingly, the two groups of children reacted differently during medial reach in the medial ankle dorsi/plantarflexion, where children with CP plantarflexed the medial ankle, but TD children dorsiflexed it.

In summary, during the hand acceleration phase of reaching, the two groups of children were found to perform different angular motions with certain movement planes, such as the transverse plane, at medial reach. Children with CP extended and side-bent the trunk toward the medial side with decreased trunk rotation to that side while also

tilting the pelvis down toward the medial side during medial reach. However, TD children rotated the trunk and pelvis (increased range of rotation) toward the medial side majorly to manage the medial reach task. For lateral reaching, children with CP rotated the pelvis toward the lateral side with no obvious trunk side-bending, but the TD children stabilized the pelvis (no pelvis rotation) with subtle side-bending movement for trunk adjustment.

For lower extremity control, two groups of children performed different movement patterns at medial reach. Children with CP adducted the lateral hip (moving toward the reach side) with ankle plantarflexing of the medial ankle, whereas TD children abducted the lateral hip (moving away from the reach side) with ankle dorsiflexing of the medial ankle.

*3.3. Angular Changes in Axial and Lower Extremity Components during the Hand Deceleration Phase*

For head movement, a group effect existed in three planes of movement. Children with CP flexed the head, but TD children attempted to extend the head at this phase (group effect: F3.357, df = 1, $p$ = 0.035). In addition, TD children demonstrated the greatest range of head extension at lateral reach (ant vs. lat40: $p$ = 0.006; lat40 vs. med40: $p$ = 0.016, Table 3). In general, children seemed to side-bend and rotate the head toward the lateral side during the hand deceleration phase of reaching; however, during the medial reach, children with CP side-bent their head in the lateral direction to a greater degree than did the TD children. For head rotation, children with CP rotated their head toward the lateral direction to a lesser degree than did the TD children. For the task effect, children side-bent their head toward the lateral direction to a greater degree for medial reach than for anterior or lateral reaches. The children decreased the range of head side-bending to the lateral side at lateral reach compared to anterior and medial reaches. For head rotation, the children increased the range of head rotation to the lateral side when reaching medially compared to when reaching anteriorly or medially. No interactive effect between group or task was found.

For trunk movement, only task effects were found in all three planes of movement. The subjects tended to decrease the range of trunk extension movement during medial reach compared to the other two reach directions (F = 5.159, df = 2, $p$ = 0.013). For trunk side-bending (F = 54.537, df = 2, $p$ < 0.001), children moved their trunk by side-bending to the medial side while reaching laterally, but would move their trunk by side-bending to the lateral side when reaching medially. When they reached anteriorly, no obvious trunk movement was noted. For trunk rotation (F = 33.380, df = 2, $p$ < 0.001), the greatest range of trunk rotation toward the medial side was found during medial reach. Again, all children decreased the moving range of trunk rotation toward the medial side at lateral reach.

For pelvis movement, group effects were found in sagittal and transverse planes of movement. Task effects existed in all planes of movement. A significant interactive effect in pelvis rotation was seen. Children with CP demonstrated fewer degrees of pelvis anterior tilting at lateral and medial reaches than did TD children (group effect: F = 6.824, *df* = 1, $p$ = 0.002). Children with CP demonstrated greater pelvis rotation toward the lateral side at lateral reach than did TD children (group effect: F = 8.221, df = 1, $p$ = 0.001); however, they showed fewer degrees of pelvis rotation toward the medial side at medial reach than did TD children. The children decreased the range of anterior tilt more for medial reach than for anterior or lateral reaches (task effect: F = 6.996, df = 2, $p$ = 0.004).

In the other two planes of movement, the children showed different movement adjustment according to the task condition, tilting the pelvis down to the lateral side to a greater degree when reaching laterally than in the other two directions, but further adjusted the pelvis tilting down to the medial side when reaching medially (task effect: F = 11.692, df = 2, $p$ < 0.001). However, children with CP reacted differently than TD children (interaction effect: F = 10.592, df = 2, $p$ < 0.001), performing measurable degrees of pelvis rotation toward the lateral side while reaching laterally, but TD children did not rotate the pelvis. When reaching medially, children with CP rotated the pelvis toward the medial side to a lesser degree than did TD children.

**Table 3.** Mean and standard deviation of dependent variables of angular changes in axial and lower extremity components during the hand deceleration phase.

| Task<br>Variable (°) | CP | | | TD | | |
|---|---|---|---|---|---|---|
| | ant | lat40 | med40 | ant | lat40 | med40 |
| Head extension/flexion | −4.73 (7.54) | −5.15 (6.81) | −4.19 (8.86) | 1.41 (3.02) | −1.67 (5.59) | 3.11 (6.93) |
| Head side bend to lat/med | 5.63 (5.23) | 2.60 (5.21) | 13.11 (9.00) | 2.93 (2.98) | 1.24 (3.73) | 4.86 (5.04) |
| Head rotation to lat/med | 5.27 (4.96) | 4.21 (6.34) | 6.84 (5.64) | 9.79 (3.45) | 8.47 (4.27) | 13.80 (4.37) |
| Trunk extension/flexion | 9.48 (4.49) | 8.78 (6.02) | 6.79 (5.97) | 8.10 (3.78) | 10.18 (4.74) | 6.08 (4.52) |
| Trunk side bend to lat/med | −1.76 (2.44) | −6.19 (3.85) | 3.65 (5.18) | −0.82 (2.57) | −5.51 (3.29) | 4.36 (5.70) |
| Trunk rotation to lat/med | −8.67 (4.46) | −5.91 (5.05) | −13.39 (4.20) | −11.71 (3.63) | −8.36 (3.78) | −16.80 (3.86) |
| Pelvis post/ant tilt | −2.55 (4.13) | −1.53 (2.62) | −0.36 (3.16) | −3.96 (3.24) | −5.37 (3.75) | −2.92 (3.18) |
| Pelvis tilt down to lat/med | 1.23 (3.79) | 2.54 (3.78) | −0.17 (2.91) | −0.29 (2.81) | 2.63 (4.32) | −2.50 (1.59) |
| Pelvis rotation to lat/med | −2.21 (2.35) | 1.84 (2.47) | −4.91 (2.97) | −2.64 (1.46) | 0.09 (1.55) | −10.46 (3.54) |
| Lateral hip flexion/extension | 4.21 (3.92) | 4.41 (3.23) | 1.03 (3.37) | 4.99 (2.90) | 6.48 (3.13) | 0.70 (2.62) |
| Lateral hip adduction/abduction | 0.79 (1.73) | 1.18 (4.83) | −0.37 (1.66) | −2.40 (1.74) | −0.27 (1.54) | −11.40 (5.76) |
| Medial hip flexion/extension | 2.97 (3.97) | 1.69 (3.38) | 1.40 (3.87) | 4.36 (2.39) | 2.85 (1.75) | 3.87 (2.55) |
| Medial hip adduction/abduction | 2.36 (2.74) | 0.27 (3.56) | 0.72 (2.53) | 2.33 (1.39) | −0.15 (2.51) | 8.94 (3.32) |
| Lateral ankle dorsi/plantarflexion | −0.01 (2.91) | −1.36 (2.08) | −0.21 (3.55) | 0.82 (1.27) | −0.33 (1.27) | 2.50 (2.04) |
| Lateral ankle supination/pronation | 1.13 (3.65) | 2.58 (3.61) | 1.22 (3.35) | −1.30 (1.66) | −2.18 (1.84) | −1.01 (1.85) |
| Lateral ankle abduction/adduction | 1.82 (3.35) | 0.55 (3.60) | 2.66 (3.34) | 0.81 (1.16) | −0.31 (1.74) | 1.76 (2.41) |
| Medial ankle dorsi/plantarflexion | 0.87 (2.64) | 0.75 (1.78) | −0.01 (3.19) | −0.23 (1.20) | −0.83 (1.49) | 0.62 (1.57) |
| Medial ankle supination/pronation | 0.42 (3.24) | −0.07 (2.78) | −1.11 (2.51) | 0.14 (1.80) | −0.19 (1.66) | −0.08 (2.65) |
| Medial ankle abduction/adduction | −1.42 (2.19) | 2.06 (3.24) | −2.29 (3.90) | 0.80 (1.57) | 0.96 (1.87) | −0.01 (2.72) |

Note: lat: lateral side (the same side as lateral reach; the same side as the preferred hand); med: medial side (the same side as medial reach; the same side as the non-preferred hand).

For hip movement, group and interactive effects existed in hip adduction/abduction (group effect: F = 14.343 and 17.558, respectively, df = 2, $p < 0.001$: interaction effect: F = 14.623 and 24.515, respectively, df = 2, $p < 0.001$). Children with CP did not move the lateral hip when performing anterior or medial reaches, but TD children abducted the lateral hip. In addition, children with CP did not move the medial hip, but TD children adducted the medial hip when reaching medially. Children with CP did not adjust lateral hip motion as did TD children when reaching in different directions. TD children demonstrated several degrees of hip abduction while reaching anteriorly, and kept the hip motionless and further abducted the lateral hip while reaching medially (Table 3).

For ankle movement, group effects existed in lateral ankle dorsi/plantarflexion (F = 3.165, df = 1, $p = 0.042$), supination/pronation (F = 6.747, df = 1, $p = 0.002$), and medial ankle abduction/adduction (F = 5.898, df = 1, $p = 0.003$). A task effect was revealed in ankle abduction/adduction motion for both legs (F = 5.923 and 9.470, respectively, df = 2, $p = 0.008$ and 0.001, respectively), whereas an interactive effect was revealed in lateral ankle supination/pronation (F = 3.568, df = 2, $p = 0.043$) and medial ankle abduction/adduction (F = 6.847, df = 2, $p = 0.004$). Children with CP generally supinated the lateral ankle while reaching compared to TD children, who pronated the lateral ankle. Children with CP further supinated the lateral ankle when reaching laterally, but TD children pronated it to a greater degree. It seemed TD children used a pronation motion to stabilize the lower extremity. Children with CP adducted the medial ankle while reaching anteriorly and medially and abducted it when reaching laterally, but TD children showed no ankle motion in any of the three reach directions.

In summary, no interactive effect was found in head or trunk movement during the hand deceleration phase of reaching. Different angular patterns were observed in pelvis, hip, and ankle motion between groups. Children with CP rotated the pelvis toward the lateral side when performing the movement of lateral ankle supination when reaching laterally. TD children did not rotate the pelvis, but abducted the hip laterally when performing lateral ankle pronation. During medial reach, all children rotated the pelvis toward the medial side, but children with CP performed fewer degrees of rotation than did children with TD. Children with CP did not adjust subtle lateral hip abduction as TD children did; however, children with CP adducted the medial ankle, but TD children kept it motionless during hand deceleration for medial reach.

## 4. Discussion

Our two reported studies have found that children with CP react differently than TD children during reaching tasks, especially for medial reaching [24,25]. In this detailed kinematic analysis, while reaching in different directions, various angular changes of joint movement were noticed between groups. Children with CP perform different forms of angular patterns of head, trunk, and pelvis segments to manage different tasks than TD children.

For postural preparation 300 ms before reach began, children with CP oriented their head and trunk toward the reaching direction in advance, but TD children did not. Children with CP rotated their head and trunk to a greater degree than did TD children to manage the different reach directions; therefore, they might not need to deal with postural adjustment (rotating the body toward the reach direction), postural stability, or hand reach (controlling the upper extremities) at the same time. Shumway-Cook and Woollacott (2007) mentioned that postural control could be categorized into two parts: postural orientation and postural stability [29]. This study demonstrated that children with CP chose to align the head and trunk first in order to decrease the demand of the task (needing to re-orient the body and stabilize it at the same time during the task). This result suggests that children with CP perceive their deficit and choose an alternative way to adapt their ability in order to complete the task safely. In Pavão's study, the researchers also found similar results. Children with CP (mean age: 9.5 years) who encountered difficulty in controlling reach and postural stability at the same time found much more advantage in external trunk support than children with CP, who demonstrated fewer problems controlling dual tasks [14].

Furthermore, when the target is located in different directions, such as on the medial side of the trunk, subjects have to deal with the Coriolis force created with trunk rotation, and researchers have found that people create more flexor movement of the upper extremities to counterbalance the Coriolis effect if they reach medially compared to other directions [30]. It is possible that children with CP prepare alignment before reaching begins in order to decrease the need of the upper extremities to deal with the effect of the Coriolis force, since children with CP have deficits in recruiting motor movement efficiently or regulating force in advance [15,16]. In our study, we consistently found that children with CP reacted differently than TD children during medial reaching.

A previous investigation reported that children with CP spent more time on reaching than returning (normalized movement time relative to the reach cycle: 54.2% vs. 45.8%, respectively), but TD children did not show this strategy (47% vs. 53%, respectively) during medial reach. In addition, children with CP demonstrated a larger straightness ratio (SR) of the hand-reach path (the larger the SR, the less efficient the reach) at medial reach, but TD children maintained rather stable SR (close to 1) [24]. Besides, it was also noticed that children with CP demonstrated different postural adjustments than TD children [25]. During acceleration and deceleration phases of medial reach, a greater proportion of children with CP used more apparent pushing or braking force for balancing than did TD children, so these kinematic results further confirm this different phenomenon observed between the groups.

In the acceleration and deceleration phases of medial reach, children with CP rotated their trunk and pelvis to a lesser degree than did TD children (Tables 2 and 3). Children with CP also did not efficiently rotate the trunk toward the medial side as did those with TD (significant interactive effect of trunk rotation; Table 2), so they chose to rotate the body in advance, thereby decreasing the need for trunk and pelvis rotation during reaching. It is therefore evident that children with CP use an alternative strategy to manage a specific task demand while considering their characteristics (e.g., insufficient postural control). Newell (1986) mentioned that the interplay between individual and task constraints affects the motor solution chosen, and other previous studies have also found similar results [5,31].

In the angular movement of the pelvis and lower extremities, this study found that children with CP and TD reacted differently when managing different task constraints. For lateral reach, children with CP chose to align the medial hip laterally before beginning reaching, and as the reach started, they rotated the pelvis laterally and supinated the lateral ankle. However, TD children aligned the medial hip medially before beginning reaching, and as the lateral reach started, they kept the pelvis stable, abducted the lateral hip, and pronated the lateral ankle. For medial reach, children with CP chose to align the medial hip in a neutral position before the reach, and then after the reach began, they decreased the use of pelvis rotation to the lateral side but increased the use of tilting the pelvis down to the medial side while adducting the lateral hip and plantarflexing the medial ankle. However, TD children aligned the medial hip medially before reaching, and then after the reach began, abducted the lateral hip and adducted the medial movement with increased pelvis rotation to the medial side while also dorsiflexing the medial ankle. It seemed that the lower extremities of children with CP were not functionally involved in postural control during lateral and medial reaches, and the strategy chosen by them seemed to threaten their postural stability. This was evident in the previous report [25].

The difficulty of medial reaching being performed in children with CP was observed consistently in our study. These children showed a greater straightness ratio of hand reach than did TD children at medial reach [24]. Children with CP made more exaggerated postural adjustments than TD children in COP performance and force regulation [25]. In this research report, children with CP perceived the insufficiency of their trunk control and came up with an alternative strategy, using increased shoulder adduction with decreased trunk and pelvis rotation while recruiting increased trunk side-bending movement and attempting to stabilize the medial leg with increased medial ankle adduction.

This finding showed some similarities with previous studies where children with CP recruited different degrees of freedom of motion to deal with the task in order to finish the task successfully [21,32]; however, the way they chose to manage the task appeared likely to be detrimental to their postural stability while performing the task. This information could be very important for physical therapy intervention.

In summary, the analysis of postural alignment and the angular movement of body segments revealed that children with CP could perceive their ability challenges and decide to prepare their alignment in advance so they could adapt to the task and decrease the consequent postural challenge by coming up with different motor patterns than TD children. However, the alternative strategies chosen by children with CP might not be the most efficient way to perform the task safely. Finally, these findings have disclosed an issue: Children with insufficient postural control ability (e.g., CP) have trouble in dealing with medial reach. Further studies should investigate this finding in depth and generate such valuable information for clinical use. Last, this research report is part of results based on a PhD thesis [33].

**Author Contributions:** Conceptualization: Y.-H.J. and R.-J.C.; methodology: Y.-H.J. and R.-J.C.; formal analysis and investigation: Y.-H.J.; writing—original draft preparation: Y.-H.J.; writing—review and editing: Y.-H.J. and R.-J.C.; funding acquisition: Y.-H.J.; resources: Y.-H.J.; supervision: Y.-H.J. and R.-J.C. All authors have read and agreed to the published version of the manuscript.

**Funding:** This research was financially supported by the National Science Council, Taiwan (NSC 94-2314-B-037-069).

**Institutional Review Board Statement:** The study was conducted according to the guidelines of the Declaration of Helsinki, and approved by the Institutional Review Board of Kaohsiung Medical University, Chung-Ho Memorial Hospital (KMUH-IRB-940289 and date of approval: 23 December 2005).

**Informed Consent Statement:** Informed consent was obtained from all subjects involved in the study.

**Data Availability Statement:** The datasets used and/or analyzed during the current study are available from the corresponding author on reasonable request.

**Acknowledgments:** Acknowledgements go to the participating children and their families. The authors also extend thanks for the help from the Statistical Analysis Laboratory, Department of Clinical Research, Kaohsiung Medical University, Chung-Ho Memorial Hospital.

**Conflicts of Interest:** The authors declare no conflict of interest.

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
