# Peer review of "Effects of Task Constraints on Kinematic Characteristics of Body Segments in Children with Cerebral Palsy and Children with Typical Development during Hand-Reaching"

_applsci, doi:10.3390/app112311140_

Round 1

Reviewer 1 Report

Dear Authors,

Congratulations on your work. The research you present is interesting, well described and well argued. However, modifications and improvements are necessary before its possible publication.

1. Abstract. The abstract in this publication needs to be unstructured. In addition, the use of abbreviations in the abstract is discouraged. Correct and rewrite this section appropriately.

2. Introduction. This section should be slightly abridged and some of the references used should be replaced by more recent ones. For example:

DOI 10.1016/j.gaitpost.2021.04.027
DOI 10.3390/jcm10010137
DOI 10.3390/diagnostics11010008
DOI 10.3390/ijerph17186836
DOI 10.1186/s12887-020-02073-1
DOI 10.3390/jcm9092919

All these references may be useful to contextualise the importance of lifestyle factors in the maturation of motor control during childhood.

Materials and Methods. No mention is made of the methodological design of the study or the type of sample used. The methodological design should also be present in the Abstract.

Comparisons between groups (even if they are pre-test comparisons) belong to the Results section. These data should appear at the beginning of the following section.

In addition, the subsections Data reduction and Statistical analysis should be differentiated. The statistical techniques used should at least be complemented by the calculation of effect sizes.

4. Results. The use of the first person is generally discouraged throughout the text, but even more so in this section.

5. Discussion. This section should begin by restating the objective of the study and providing an answer to it.

There are references that are not in the proper format of the Journal.

The use of few bibliographical references (and their age) in this section stands out negatively. You can take note of the references previously recommended as a complement in this section. In this way, you will be able to enrich your text.

6. Conclusions. Define this section clearly.

Kind regards.

Reviewer 2 Report

See the attached document.

Round 2

Reviewer 1 Report

Dear Authors,

Congratulations on the improvements made.

Kind regards